# Exploring the Antibacterial and Antibiofilm Efficacy of Silver Nanoparticles Biosynthesized Using *Punica granatum* Leaves

**DOI:** 10.3390/molecules26195762

**Published:** 2021-09-23

**Authors:** Monisha Singhal, Sreemoyee Chatterjee, Ajeet Kumar, Asad Syed, Ali H. Bahkali, Nidhi Gupta, Surendra Nimesh

**Affiliations:** 1Department of Biotechnology, IIS (Deemed to be University), Gurukul Marg, SFS, Mansarovar, Jaipur 302020, India; monishasinghal50@gmail.com (M.S.); sreemoyee.chatterjee@iisuniv.ac.in (S.C.); 2Department of Chemistry and Biomolecular Science, Clarkson University, Potsdam, NY 13699-5814, USA; ajeetkumar5@gmail.com; 3Department of Botany and Microbiology, College of Science, King Saud University, P.O. 2455, Riyadh 11451, Saudi Arabia; assyed@ksu.edu.sa (A.S.); abahkali@ksu.edu.sa (A.H.B.); 4Department of Biotechnology, School of Life Sciences, Central University of Rajasthan, Ajmer 305817, India

**Keywords:** *Punica granatum* leaves extract, silver nanoparticles, Box-Behnken design, antibacterial, antibiofilm

## Abstract

The current research work illustrates an economical and rapid approach towards the biogenic synthesis of silver nanoparticles using aqueous *Punica granatum* leaves extract (PGL-AgNPs). The optimization of major parameters involved in the biosynthesis process was done using Box-Behnken Design (BBD). The effects of different independent variables (parameters), namely concentration of AgNO_3_, temperature and ratio of extract to AgNO_3,_ on response viz. particle size and polydispersity index were analyzed. As a result of experiment designing, 17 reactions were generated, which were further validated experimentally. The statistical and mathematical approaches were employed on these reactions in order to interpret the relationship between the factors and responses. The biosynthesized nanoparticles were initially characterized by UV-vis spectrophotometry followed by physicochemical analysis for determination of particle size, polydispersity index and zeta potential via dynamic light scattering (DLS), SEM and EDX studies. Moreover, the determination of the functional group present in the leaves extract and PGL-AgNPs was done by FTIR. Antibacterial and antibiofilm efficacies of PGL-AgNPs against Gram-positive and Gram-negative bacteria were further determined. The physicochemical studies suggested that PGL-AgNPs were round in shape and of ~37.5 nm in size with uniform distribution. Our studies suggested that PGL-AgNPs exhibit potent antibacterial and antibiofilm properties.

## 1. Introduction

Nanotechnology deals with the manipulations of materials at molecular and sub-molecular levels. It is primarily concerned with the development of natural and artificial nanostructures. The nanoparticles explicitly lie in the size range of 1 nm to 100 nm in any one dimension and possess features that vary from their bulk counterpart. The properties of nanoparticles tend to change due to the reduction in size and structure of the materials to the atomic level. This has made them compatible for various applications such as diagnostic tools, vehicles for drug loading, biosensors, sunscreens formulation, antimicrobial therapeutics in the form of bandages and disinfectants and as a catalyst. These nanoparticles, therefore, offer greater efficiency in the manufacturing process as the use of toxic materials can be reduced [1,2]. Metallic nanoparticles, specifically silver nanoparticles (AgNPs), have been utilized in various applications due to their unique broad-spectrum antibacterial and antifungal activity. These inorganic nanoparticles exhibit low toxicity against mammalian cells as compared to their bulkier salt form [3]. AgNPs are well known to show their effectiveness against multidrug-resistant bacteria.

Various chemical and physical methods are available for the synthesis of these AgNPs that are capable of producing well-defined, small-sized nanoparticles. However, all these methods are associated with certain limitations, as these are very costly, energy-consuming and utilize hazardous chemicals. Aside from these limitations, low stability and aggregation of the formulated nanoparticles is another challenge [4]. To combat these issues, researchers are now moving towards the biological synthesis of these AgNPs, including microbial (bacteria-, fungi-based) synthesis and green (plant-based) synthesis [5,6,7].

Plants extract have the ability to reduce and act as a capping agent during nanoparticles synthesis and are considered advantageous over microbial synthesis owing to its less cumbersome process [8]. Thus, the elaborated process of culture and maintenance of microorganisms can be avoided, and plant extract mediated synthesis can easily be amplified for a larger scale [8]. Different types and parts of a plant such as *Prosopis juliflora* bark, *Prosopis juliflora* leaf, *Cicer arietinum* leaves, *Trigonella foenum-graecum* seed, *Terminalia chebula* fruit, *Saraca indica* flower, *Punica granatum* peel, *Acacia nilotica* leaf, etc. have been used for biosynthesis of sAgNPs [9,10,11,12,13,14,15]. Not only the plant parts but the waste from the plants have also been explored for the biosynthesis of AgNPs, such as lignin extracted from wheat straw and grape pomace from the wine industry [16,17]. Various biomolecules (such as polysaccharides, polyphenols, aldehydes, ketones, proteins/enzymes, amino acids and caffeine) present in the extracts play a vital role in the reduction of metal ions and stabilize the nanoparticles [18]. *Punica granatum* (common name, Pomegranate) is a small fruit-bearing deciduous shrub that belongs to the *Punicaceae* family. *P. granatum* leaves contain tannins, flavones, apigenin, luteolin, glycosides which enables it to have a wide range of potential health benefits. It has been shown to aid in digestion and treat certain infections and illnesses. In a recent study, the leaves of *P. granatum* were used for the biosynthesis of AgNPs and explored for their antidiabetic, anticancer and antibacterial potential [19]. In another study, the peel waste of *P. granatum* was employed for the biosynthesis of AgNPs and the antibacterial potential was explored against both Gram-negative and Gram-positive bacteria along with evaluation of its cytotoxicity against a colon cancer cell line, RKO [20]. A further study reported the use of Punicalagin, found in *P. granatum* husk, for the synthesis of AgNPs and exploration of its antioxidant, antibacterial and anticancer potential against HepG2 cancer cells [21].

Though few studies have reported the use of *P. granatum* leaves, peel or husk for the biosynthesis of AgNPs, a systematic study that examines the influence of various parameters that dictates the size of AgNPs, such as concentration of AgNO_3_, temperature, the ratio of extract to AgNO_3_, is lacking. The present study was designed to critically investigate the influence of several parameters determining the size of nanoparticles (PGL-AgNPs), such as concentration of AgNO_3_, temperature, the ratio of extract to AgNO_3_. The novelty of the present work relies on the application of the Box-Behnken design (BBD) for the optimization of these parameters in the biosynthesis process. The biosynthesis of AgNPs from AgNO_3_ was observed using UV-vis spectroscopy. Physicochemical characterization of PGL-AgNPs in terms of size and size distribution along with zeta potential was done on DLS, SEM followed by Energy-dispersive X-ray spectroscopy (EDX), FTIR and XRD. FTIR was further explored to identify the modulation in the bacterial biomolecules that could occur due to exposure to biosynthesized AgNPs. The antibacterial activity of PGL-AgNPs was studied against *Bacillus subtilis*, *Staphylococcus aureus*, *Pseudomonas aeruginosa*, *Escherichia coli* and *Proteus vulgaris.* Further, the antibacterial mechanism of PGL-AgNPs was deciphered using SEM. Finally, the antibiofilm efficacy was also explored in two biofilm-forming bacteria, i.e., *P. aeruginosa* and *S. aureus*.

## 2. Results and Discussion

### 2.1. Biosynthesis of PGL-AgNPs

The leaves of *P. granatum* are known to contain several tannins, flavones, apigenin, luteolin, glycosides, etc., that could apparently assist in the formulation of AgNPs. The AgNPs genesis was marked by the steady transition in the color of the reaction mixture from greenish-yellow to dark brown with peculiar surface plasmon resonance (SPR) peak around 420 nm, further confirming its formation (Figure 1). Arya et al. have suggested a change in color from light yellow to dark brown due to the formation of AgNPs using *Prosopis juliflora* leaf extract [14]. In another study, Ahmed et al. have also shown color transformation in the case of the colorless AgNO_3_ solution to dark brown as a result of the AgNPs formation where *Terminalia arjuna* bark extract was employed [22]. In one of the studies, Arya et al. reported the optimum synthesis of AgNPs using *Prosopis juliflora* leaf extract at 25 °C, where 0.5 mL of extract was used to reduce 9.5 mL of AgNO_3_ (1 mM) when the reaction was sustained for 40 min [14]. In another study, Veisi et al. used aqueous leaf extract of *Thymbraspicata* and reported that for an optimum synthesis, 30 min were required for the entire reduction of AgNO_3_ [23]. Biosynthesis of AgNPs using *Garcinia mangostana* leaf extract was also tested for its antimicrobial activities [24]. Chand et al. reported a study where they biosynthesized AgNPs using onion (O), tomato (T), acacia catechu (C) alone and a mixture of all three (COT) extracts [25].

### 2.2. Box-Behnken Design for Optimization of Parameters for Biosynthesis of PGL-AgNPs

Optimization of parameters for AgNPs biosynthesis is essential in order to improve the stability of the prepared nanoparticles along with minimal variability in terms of size. The optimization also facilitates enhancement in the nanoparticles yields that would be one essential prerequisite in the case of bulk synthesis. The systematic optimization of parameters for the synthesis of PGL-AgNPs was conducted through BBD by Design Expert version 12.0 (Stat-Ease Inc., Minneapolis, MN, USA). In this study, 17 trial runs were suggested by BBD where the concentration of AgNO_3_ (mM), reaction temperature (°C) and the ratio of extract to AgNO_3_ (µL) were picked up as independent factors. These 17 reactions were validated experimentally, and reactions were analyzed for size on DLS. The data procured were further analyzed using a quadratic polynomial model, and model diagnostic plots were generated to evaluate if the selected model fitted well for the given data. The model diagnostic plots for response particle size are shown in Figure 2A–C. Here, we obtained a linear predicted vs. actual graph indicating that the predicted values were very much in close proximity with the actual ones. The effect of all the selected factors at a particular point in the experiment design was predicted by a perturbation plot. The graph shows a slight curvature for factor C (i.e., temperature), indicating the response’s sensitiveness to that particular variable. While other two factors (i.e., A-AgNO_3_ concentration and B- extract to AgNO_3_ ratio) were relatively flat, indicating no such effects on response when the variables fluctuate. Also, there was no significant mean effect between AgNO_3_ concentration and the extract to AgNO_3_ ratio, as the green and red mean lines were parallel to each other, as displayed in the interaction plot.

Similarly, model diagnostic plots of response polydispersity index (PDI) were generated. Figure 2D–F illustrates predicted vs. actual plots, where experimental values obtained were quite close with the predicted values. The perturbation plot was found to have deviated much, as all the factors showed curvatures from the desired point, which means a change in any factor affects the response to a certain extent. The interaction plot revealed a significant mean effect among two selected variables as the mean effect lines intersect with each other.

### 2.3. Factor-Response Association and Response Surface Analysis

Box and Wilson, in 1951, introduced response surface methodology (RSM) as a tool that combines mathematical and statistical approaches to optimize variables or factors for a given model system [26]. Here the existence of all the possible interactions amongst the three independent variables with their impact on responses was evaluated by 2D contour plots and 3D response surface plots. In Figure 3A,D, the effect of AgNO_3_ concentration and extract to AgNO_3_ ratio on response particle size was determined. The 3D response surface plot depicts that there was no such association of both the variables on particle size, yet particle size slightly increases at higher levels. The 2D contour plot also predicts the maximum number of particles formation at higher levels. The link between AgNO_3_ concentration and temperature is shown in Figure 3B,E. It has been seen that when elevating the AgNO_3_ concentrations at low levels of temperature, a declining pattern of particle size was observed. Whereas at higher temperatures, an increase in AgNO_3_ concentration showed an inclination pattern of particle size. The 2D contour plot also revealed the same relationship. The relationship between extract to AgNO_3_ and temperature was depicted in Figure 3C,F. Increasing the volume of the extract at a lower range of temperatures showed a drop in the pattern of particle size, whereas when increasing the extract volume at a higher level of temperatures, an increment in particle size was observed.

Similarly, the influence of all three variables on the response polydispersity index was evaluated. In Figure 4A,D, the 3D surface plot showed a complex interaction between AgNO_3_ concentration and extract to AgNO_3_ ratio where the increased concentration of AgNO_3_ at lower levels of extract volume, a declining pattern of PDI was observed. However, rising the concentration of AgNO_3_ at a high level of extract volume, an inclining pattern of PDI was observed. Figure 4B,E predicts the effect of AgNO_3_ concentration and temperature on PDI. an increase in the concentration of AgNO_3_ at a lower level of temperature exhibited a low range of PDI, while at a higher level of temperature, a low range of PDI was observed. In Figure 4C,F, increasing extract volume at low-temperature levels, PDI tends to increase, while increasing extract volume at high-temperature levels, a declining pattern of PDI was seen.

### 2.4. Studies for Optimized PGL-AgNPs

The optimization of parameters for PGL-AgNPs biosynthesis was attained by numerical optimization through BBD. Once the experimental validation of 17 reactions was done, the obtained data for responses—particle size and PDI—were subjected to numerical optimization. Herein, the data were adjusted in the desired range, and an overlay plot was generated to depict the desired conditions for biosynthesis purposes. In Figure 5, the yellow region predicted the optimized area for the biosynthesis of silver nanoparticles. The plot depicts that desired range of PDI, i.e., below 0.3, obtained at a higher concentration of AgNO_3_ while keeping extract volume at the lowest range. The flagged point describes the optimized parameters for the same. As a result, the uniformly distributed nanoparticles can be biosynthesized when the concentration of AgNO_3_ is 1.5 mM, with extract volume of 55.55 µL taken and the reaction carried out at 31.4 °C continued for 15 min.

### 2.5. Physicochemical Characterization of PGL-AgNPs

UV-vis spectroscopy is routinely employed for the characterization of metallic nanoparticles, as they display SPR in the visible light spectrum. SPR is an optical approach widely practiced for the determination of molecular interactions in nanoparticles. It is based upon the cumulative oscillation of free electrons found on the surface of metal nanoparticles that determines the SPR bands. The narrow size distribution of AgNPs is typically depicted by a sharp peak. This peak may vary with the type, size and shape of nanoparticles. The size of particles and dielectric medium are two main factors; broad peak and prominently small peak depict the vast and mono-dispersed distribution of nanoparticles, respectively [27,28]. In the event of spherical nanoparticles, a single SPR peak is likely to appear, while more than two peaks appear in the event of anisotropic nanoparticles [29].

In the present study, the reaction mixture showed an absorbance peak at around 420 nm, which is distinctive of AgNPs due to its SPR absorption band. Size determination studies done on SEM showed that PGL-AgNPs were spherical with smooth surface morphology with an average size of ~37.5 nm (Figure 6A). EDX analysis of biosynthesized nanoparticles revealed the presence of silver atoms (Figure 6B). Further, the zeta potential was found to be −34 mV upon analysis of samples on Zetasizer (Figure 6D). An XRD study of the PGL-AgNPs disclosed prominent peaks corresponding to (111), (200), (220) and (311) Bragg’s reflection based upon the fcc structure of AgNPs (Figure 6C). AgNPs synthesized using *Rheum turkestanicum* shoots extract by Yazdi et al. also reported the four Bragg diffraction values 38.2, 44.6, 224 64.6 and 77.5 at 2θ representing the (111), (200), (220) and (311) set of lattice plane, respectively [30]. The synthesis of AgNPs is generally illustrated by the expansion of Bragg’s peaks. The XRD data received for PGL-AgNPs was compared with the database of Joint Committee on Powder Diffraction Standards (JCPDS) file No. 04-0783. Further, the stability of AgNPs was checked in terms of analysis on UV-vis spectrophotometer after an interval of 30 days of storage at room temperature. No comparative shift in the spectrum was observed for stored AgNPs in contrast with the freshly prepared AgNPs.

### 2.6. Fourier-Transform Infrared Spectroscopy

The FTIR spectrum of *P. granatum* leaves extract and PGL-AgNPs was recorded within the wavenumber range of 4000–400 cm^−1^ region. The AgNPs showed major peaks at 1623 cm^−1^ (N-H bend), 1191 cm^−1^ (C-O stretch), 1025.5 cm^−1^ (C-N stretching) and 756 cm^−1^ (C-Cl stretching), suggesting the relationship of primary amines, aliphatic amines and alkyl halides of the constituting biomolecules present in the extract (Figure 7). N-H stretching comparable to primary and secondary amines was found in both extract and AgNPs, which implies that they might be liable for capping and stabilization of PGL-AgNPs. In the PGL extract, the band at 3275 cm^−1^ was attributed to the polyphenols, while in the PGL-AgNPs, the vibration band moved toward the shorter frequency of 3219 cm^−1^, suggesting the implication of functional groups associated with polyphenols as reducing agents during biosynthesis of AgNPs [31]. The band at 2933 cm^−1^ due to O-H stretching vibration, band at 1705 cm^−1^ due to C=O stretching vibration, the band at 1327 cm^−1^ due to C-O stretching vibration and the band at 1444 cm^−1^ due to O-H bending vibration visible in the PGL extract were assigned to the phenolic acids, and also moved to the shorter frequency in the PGL-AgNPs, substantiating the implication of phenolic acids as reducing and capping agents [32]. The band at 1029 cm^−1^ could be attributed to C-OH stretching vibration that arises due to the presence of polysaccharides.

### 2.7. Antibacterial Studies

Ag^+^ and Ag salts containing silver have been used as antimicrobial agents for decades owing to their inherent microbial growth inhibition potential. The application of Ag^+^ or salts is limited due to the interfering effects of salts and the requirement of the uninterrupted release of sufficient concentration of Ag ion for the antimicrobial effect. These drawbacks can be surmounted by the use of AgNPs, as silver possesses tremendous antimicrobial properties compared to other salts. Due to the greater surface area of AgNPs, it would have superior binding efficiency with the microorganisms [33]. AgNPs have been suggested to exhibit an antibacterial response through multiple mechanisms. One of the proposed mechanisms suggests an attachment of AgNPs to the surface of the cell membrane of the bacterium, perturbing its function, invaginating the bacterium, followed by the release of silver ions that leads to the effective antibacterial activity against Gram-negative bacteria [34]. Another mechanism suggests anchorage to the cell wall followed by penetration and ultimately resulting in cell death [35,36]. Yet another mechanism proposes the release of free radicals produced by AgNPs that lead to damage to the cell membrane, enhance membrane permeability, resulting in cell death [37]. However, the development of economically effective strategies for the production of AgNPs is highly desirable. AgNPs have shown immense potential as antibacterial agents both against Gram-negative and Gram-positive bacteria [38,39,40].

In the present study, the antibacterial efficacy of PGL-AgNPs was assessed against both Gram-negative and Gram-positive bacteria using disc diffusion assay where ampicillin was employed as a positive control because it is effective against a broad spectrum of Gram-positive and Gram-negative bacteria as it targets the synthesis of the bacterial cell wall [41]. An increment in the zone of inhibition was found when the concentrations of PGL-AgNPs were raised from 50 μg/mL to 200 μg/mL (Figure 8). The outcomes indicate that the biosynthesized PGL-AgNPs were quite effective against both the studied Gram-positive and Gram-negative bacteria (Appendix A, see Appendix A). Sondi et al. also reported inhibition of *E. coli* with a hike in AgNPs concentration and the initial number of cells used for the tests [35]. AgNPs of 50–60 μg/cm^3^ concentration exhibited 100% inhibition of growth, and the cells were impaired as a result of the development of cracks in the bacterial cell wall [35]. Potent bacterial growth inhibition of *P. aeruginosa* and *E. coli* was observed at concentrations as low as 25 and 50 μL using *P. granatum* leaves extract biosynthesized AgNPs [20].

MIC was further done to detect the minimum concentration of AgNPs that can inhibit the growth of the bacterial culture. It was accomplished that various bacterial cultures growth can be rendered by PGL-AgNPs at a concentration range of 0.050–0.125 mg/mL. Growth of Gram-negative bacteria, namely, *E. coli*, *P. aeruginosa* and *P. vulgaris*, were observed to be inhibited at low concentrations of AgNPs compared to Gram-positive bacteria. This could be due to the presence of a sturdy cell wall in the latter, which makes the accessibility of silver nanoparticles into the cell difficult. Several research groups have reported this sort of difference in the MIC of green synthesized AgNPs (Table 1) [42,43,44,45]. This could be due to several factors that govern the efficacy of nanoparticles including, size, surface-to-volume ratio, the bioactive molecules that help in the reduction and synthesis of nanoparticles. Nonetheless, in this study, PGL-AgNPs were still found to reduce the growth of Gram-positive bacterial cultures, referring to many beneficial aspects of silver nanoparticles as efficient antibacterial agents.

### 2.8. Mechanism of Antibacterial Action of PGL-AgNPs

The study of the mechanism of antibacterial action of biosynthesized AgNPs was done on Gram-positive bacterium—*S. aureus*—as the thickness of the cell wall is much greater than the Gram-negative species. The biosynthesized AgNPs can easily invade the cell wall of Gram-negative bacteria if the nanoparticles are capable of causing damage to Gram-positive bacteria [46]. The thickness of peptidoglycan, a key component of the bacterial cell wall, plays an important role in bacterial invasion. In Gram-positive bacteria, ~30 nm thick layer of peptidoglycan is present, while only 3–4 nm thick layers are present in Gram-negative bacteria such as *E. coli*. The exposure of AgNPs to bacteria leads to their attachment to the bacterial cell wall, which is a pivotal step in order to degrade bacteria. In this study, changes in the bacterial membrane after the treatment of PGL-AgNPs were seen by performing FTIR spectroscopy. This technique was explored to determine the structural modifications by evaluating the changes in functional groups found in the bacterial membrane. Transmittance FTIR is quite popular for the purpose of perceiving the interactions between silver nanoparticles and bacterial cells [47]. Figure 9A,B illustrates the difference between membranes of *S. aureus* and PGL-AgNPs treated *S. aureus.* The structural modifications after the treatment were predominantly classified into four regions viz. lipids (3100–2800 cm^−1^), proteins and peptides (1800–1500 cm^−1^), carbohydrates (1200–900 cm^−1^) and nucleic acids (1200–600 cm^−1^) [48,49,50]. The FTIR spectrum of treated bacteria revealed an absence of peaks corresponding to the nucleic acid region, i.e., 1200–600 cm^−1^. Only one peak observed in this region, which was at 1077.11 cm^−1^, falls under the category of C-O stretching of primary alcohols.

SEM scanning was also done to anticipate and compare the surface modifications in treated as well as control bacteria. In the case of control (Figure 10A), a smooth and round surface of the bacterial membrane was observed, while in the case of treated bacteria (Figure 10B), pores were observed in the cellular membrane and contents seemed to have leaked out. The reason being silver nanoparticles bind with the surface of the bacterial membrane and causes various physical changes like membrane disturbances due to which content of cells leaked out.

### 2.9. Antibiofilm Studies

The Congo red assay was performed to detect the biofilm inhibition efficacy of biosynthesized PGL-AgNPs. Biofilms facilitate the bacteria’s survival in hostile conditions within or outside the host and are considered to be one of the reasons for chronic or persistent infections [51]. The Congo red dye labels the exopolysaccharides (EPS) secreted by the bacteria that are further utilized for the formation of biofilms [52]. The CRA method is fast, reproducible and the outcomes can be interpreted on the basis of the color of the colony obtained. The appearance of black-colored colonies exhibits biofilm production, whereas the appearance of red colonies displays biofilm inhibition. Herein, two different strains, i.e., *P. aeruginosa* and *S. aureus*, were exposed to biosynthesized PGL-AgNPs, and their biofilm formation capability was assessed. As evident from (Figure 11B,D), the red color appeared due to inhibition of biofilm formation due to exposure of both strains to PGL-AgNPs. In contrast, when the two strains were not treated with PGL-AgNPs, they formed biofilm, as evident from the appearance of black color in the region of streaking (Figure 11A,C). In one of the studies, Kalishwaralal et al. claimed inhibition of biofilm formation in *P. aeruginosa* exposed to AgNPs synthesized using *B. licheniformis* biomass, where 50 nM of AgNPs remarkably inhibited biofilm [53]. In another study, Arya et al. claimed inhibition of biofilm formation using biosynthesized AgNPs using *Prosopis juliflora* leaf extract in *B. subtilis* and *P. aeruginosa* [14].

Further, to quantitate the biofilm inhibition efficacy of PGL-AgNPs, crystal violet staining was performed (Figure 12). As exhibited by the study, more inhibition in the formation of biofilm was seen in the bacterial strains treated with PGL-AgNPs. This biofilm inhibition can be clearly visualized by observing a decrease in the intensity of color absorbed by the biofilm-forming bacteria (Figure 12A). As the AgNPs concentration was increased from 12.5 to 100 μg/mL, greater inhibition was observed in the treated bacterial strains (Figure 12B). Furthermore, the biofilm inhibition was noticed to be more against *P. aeruginosa* compared to *S. aureus* at higher concentrations of AgNPs. From these studies, it could be inferred that the biosynthesized PGL-AgNPs showed antibacterial effect via biofilm inhibition irrespective of the strain being Gram-positive or Gram-negative.

## 3. Methodology

### 3.1. Materials Required

For the biosynthesis of silver nanoparticles, silver nitrate (AgNO_3_) was purchased from Merck Limited, India. Lyophilized culture of *Escherichia coli* (MTCC 40), *Pseudomonas aeruginosa* (MTCC 424), *Proteus vulgaris* (MTCC 426), *Bacillus subtilis* (MTCC 441), *Staphylococcus aureus* (MTCC 739), were acquired from the Microbial Type Culture Collection Center (MTCC) placed at Institute of Microbial Technology (IMTECH) Chandigarh, India. Nutrient broth and Nutrient agar were procured from Hi-Media Laboratories, India.

### 3.2. Preparation of Leaves Extract

*Punica granatum* leaves were freshly collected from the neighborhood of the IIS University campus, Jaipur, Rajasthan, India 26.9124° N, 75.7873° E and identified. Fresh leaves of *P. granatum* were collected and washed twice with double distilled water to remove dirt from the surface. Clean leaves were shade dried for 15 days, followed by crushing with mortar and pestle. A 5% aqueous extract solution was prepared by boiling 5 g leaves powder in 100 mL of deionized water for 15–20 min. The solution was left at room temperature to cool. Whatman filter paper No. 1 was then used to filter the prepared extract and stored at 4 °C till further requirement.

### 3.3. Biosynthesis of Silver Nanoparticles

The biosynthesis of PGL-AgNPs was achieved by the drop-wise addition of the leaf extract into an aqueous AgNO_3_ solution with uninterrupted stirring at 600 rpm. The reaction was allowed to proceed till a color change was observed. After the stabilization of color, synthesized AgNPs were isolated by centrifugation at 10,000 rpm (ST16R, Sorvall, Thermo Scientific, Waltham, MA, USA) for 10 min. The isolated nanoparticles were washed by re-dispersing them in deionized water.

### 3.4. UV-Visible Spectroscopic Analysis

After observing the color change, the freshly biosynthesized sample was subjected to UV-visible spectrophotometric analysis. The reduction of silver ions (Ag^+^) by biomolecules in an aqueous solution was observed by taking absorbance in the range 300–700 nm using a spectrophotometer (Multiskan go, Thermo Scientific, Waltham, MA, USA).

### 3.5. Box-Behnken Design for Optimization of Parameters for Biosynthesis of PGL-AgNPs

The efficient optimization of PGL-AgNPs was achieved using the Box-Behnken design provided by Design Expert ver. 12.0 software (Stat-Ease Inc., Minneapolis, MN, USA). Different parameters that control the synthesis of AgNPs were optimized, that includes a concentration of AgNO_3_ (millimolar [mM]), reaction temperature (°C), and the ratio of the concentration of extract to AgNO_3_ (microliters [µL]). These parameters were taken as independent factors for two-level, i.e., low (−1) and high (+1), optimization process (Table 2). Based upon the preferred design, the model suggested 17 experimental trial runs for optimization of PGL-AgNPs given in Table 3. The analysis of particles size and polydispersity index (PDI) generated as responses (dependent factors) was the main reason behind this systematic optimization using BBD.

### 3.6. Physicochemical Characterization of PGL-AgNPs

Morphological and size characterization of biosynthesized PGL-AgNPs was done by Field emission-scanning electron microscopy (FE-SEM) (JEOL, 7400F, Tokyo, Japan). The existence of silver elements was proved by Energy-dispersive X-ray (EDX) analysis. Zetasizer Nano ZS with laser Doppler velocimetry was used to measure the zeta potential of the sample that was done automatically in triplicates. Also, the crystalline nature of biosynthesized AgNPs was resolved by X-ray powder diffraction (XRD). For the analysis of the functional group, desiccated samples of extract and PGL-AgNPs were assessed by Fourier-transform infrared (FTIR) spectrometer (Perkin Elmer, Yokohama, Japan). The disc of PGL-AgNPs was prepared with potassium bromide (KBr). The scanning was done in the range of 4000 to 400 cm^−1^ at 2 cm^−1^ resolution keeping the KBr pellet blank. The presence of functional groups and their interactions were detected by the FTIR spectrum.

### 3.7. Antibacterial Studies

Antibacterial activity has been tested on Gram-positive and Gram-negative bacteria, viz. *E. coli*, *S. aureus*, *P. aeruginosa*, *P. vulgaris*, *B. subtilis* by the disc diffusion method [41]. The lyophilized cultures were revived using nutrient broth and kept at 37 °C overnight to attain optimum O.D._600_ (0.4) for further use. Each plate was inoculated with 20 µL of culture and swabbed over the entire agar surface using sterilized cotton swabs. PGL-AgNPs (50 μg, 100 μg, 200 μg), *P. gratanum* extract (10%) as negative control and ampicillin (100 µg/µL) as positive control were used. Autoclaved filter paper discs loaded with varying concentrations of PGL-AgNPs were placed on the inoculated agar medium. Petri plates were then incubated overnight at 37 °C for the growth of the cultures. Zone of inhibition was deliberated in mm using a standardized scale around the disc impregnated with PGL-AgNPs, plant extract and ampicillin.

In order to determine the minimal inhibitory concentration (MIC) of biosynthesized PGL-AgNPs, the standard broth dilution method was the method of choice [54]. In the corresponding methodology, the antibacterial efficacy of PGL-AgNPs was studied against *E. coli*, *P. aeruginosa*, *P. vulgaris*, *B. subtilis* and *S. aureus* by observing the bacterial growth liquid broth medium. Silver nanoparticles were used in varying concentrations with the range of 0.010 mg/mL to 0.250 mg/mL. The microorganisms culture taken for analysis were of absorbance 0.5 at 600 nm (0.5 McFarland’s standard) and no nanoparticles. Only the broth was added to control cultures. The cultures were incubated at 37 °C for 24 h. The lowest concentration of PGL-AgNPs at which no growth was visible or observed at 600 nm was considered as MIC for PGL-AgNPs against those bacteria.

### 3.8. Mechanism of Antibacterial Action of PGL-AgNPs

With an aim to understand the antibacterial mechanism of biosynthesized PGL-AgNPs on a bacterium, FTIR and SEM were performed to observe the morphological changes in *S. aureus*. For FTIR, overnight grown PGL-AgNPs treated cultures along with proper control were taken. After the incubation, bacterial cells were collected in pellets after centrifugation and left on a desiccator to dry. The dried samples were then used to generate FTIR spectra. Similarly, overnight grown cultures, as mentioned above, were used to prepare glass slides for SEM analysis. Heat fixed smear of bacterial cultures were allowed to stain with 2.5% glutaraldehyde for 30 min, followed by washing with a phosphate buffer. The dehydration of the fixed samples was attained by drowning them in ascending percentages of an ethanol solution for 15 min each. Lastly, samples were left at 37 °C for 1 h to dehydrate completely. A thin coating of gold (<5 nm) was done before the SEM analysis [55].

### 3.9. Antibiofilm Studies

Investigation of the biofilm inhibition property of AgNPs was done using the Congo red agar method (CRA) as reported by Freeman and their group [52]. *P. aeruginosa* and *S. aureus* strains were cultivated on Luria Bertani (LB) broth provided with an incubation temperature of 37 °C for 24 h. Brain heart infusion (BHI) agar media was prepared, enriched with 5% sucrose (*w*/*v*) and 0.08% Congo red dye (*w*/*v*). All the media constituents were added and autoclaved while Congo red was separately autoclaved and then added to media at a temperature of 55 °C. Suspensions of PGL-AgNPs of increasing concentrations of 5, 10, 20, 40, 80, 100 μg/mL were made with distilled water and were spread onto separate solidified agar plates and allowed to diffuse for 6–10 min. The two-way streaking of strains was done separately and incubated at 37 °C under aerobic conditions for 24 h. The biofilm producer strains are supposed to form black colonies, while the biofilm inhibition results in red colonies.

To quantitate the biofilm inhibition efficacy of biosynthesized PGL-AgNPs, they were exposed to *P. aeruginosa* and *S. aureus*, followed by the incorporation of crystal violet dye. Moreover, 12–14 h grown cultures diluted 1:100 (*v*/*v*) in the fresh medium were transferred to wells (200 μL) of 96-well tissue culture plate (TCP) and incubated for 16 h at 37 °C without shaking. Suspension of PGL-AgNPs (10 μL) was added to each well at four different concentrations (100 μg/mL, 50 μg/mL, 25 μg/mL, 12.5 μg/mL). After measuring OD_600_, the planktonic cells were poured out, and plates were washed with 200µL of PBS (pH 7.2) to eradicate non-adherent bacteria and left to dry up for 10 min. Post drying, 200 μL of crystal violet (0.1%, *w*/*v*) was supplemented to each well and left for 10 min to stain adherent biofilms. After staining, wells were then washed with deionized water to clear surplus stain, and the dye integrated with adherent cells was dissolved by 95% ethanol. Lastly, the absorbance was taken at 595 nm on the Elisa plate reader.

The percentage inhibition of biofilm formation was obtained by calculating the absorbance discrepancy between treated and control wells. The base controls kept 200 µL of crystal violet, 200 µL of the ethanol percent reduction in growth was measured to evaluate the level of inhibition in growth of biofilm.

### 3.10. Statistical Analysis

All the experimental analyses were carried out in triplicates, with three separate experiments to describe reproducibility. All procured experimental data were provided as mean ± standard deviation (±SD).

## 4. Conclusions

In the prevailing investigation, we efficiently biosynthesized AgNPs using leaves extract of *Punica granatum*. The study revealed that optimal size distribution nanoparticles could be synthesized at 31.4 °C when the concentration of AgNO_3_ is 1.5 mM, with extract volume of 55.55 µL was taken, and the reaction continued for 15 min. Physicochemical characterization studies via SEM, DLS revealed that the PGL-AgNPs were spherical and of ~37.5 nm in size with uniform distribution and zeta potential of −34 millivolt (mV). The AgNPs exhibited effective antibacterial efficacy against the studied Gram-positive and Gram-negative bacteria as revealed by the disc diffusion method. The zone of inhibition observed for bacteria, *S. aureus*, *B. subtilis*, *P. aeruginosa*, *E. coli* and *P. vulgaris* was 14, 13, 14, 13 and 13 mm, respectively when 200 µg/mL of nanoparticles were employed. Further, the antibiofilm efficacy of PGL-AgNPs was estimated using Congo red assay and Crystal violet staining. The biofilm inhibition was higher as the concentrations of AgNPs were escalated from 12.5 to 100 μg/mL. The FT-IR and SEM studies depicted the bacterial and PGL-AgNPs interaction showing membrane disruptions in Gram-positive bacteria. These studies propose PGL-AgNPs as a potent antibacterial and antibiofilm candidate that can be explored for in vivo as well as therapeutic applications.

## Figures and Tables

**Figure 1 molecules-26-05762-f001:**
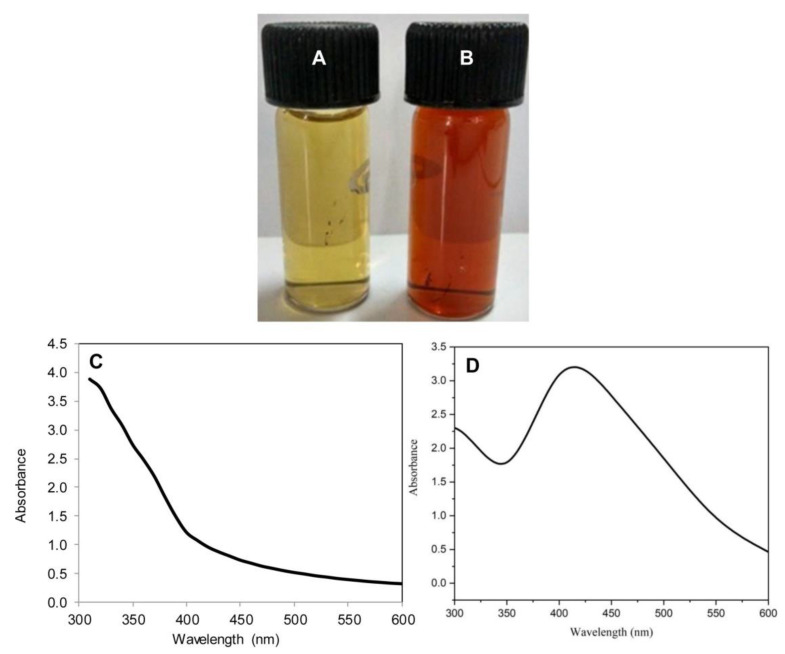
Color change showing biosynthesis of PGL-AgNPs. (**A**) PGL leaves extract, (**B**) PGL-AgNPs resuspended in double-distilled water, (**C**) UV-vis spectrum of PGL leaves extract and (**D**) UV-vis spectrum of synthesized PGL-AgNPs with a peak around 420 nm.

**Figure 2 molecules-26-05762-f002:**
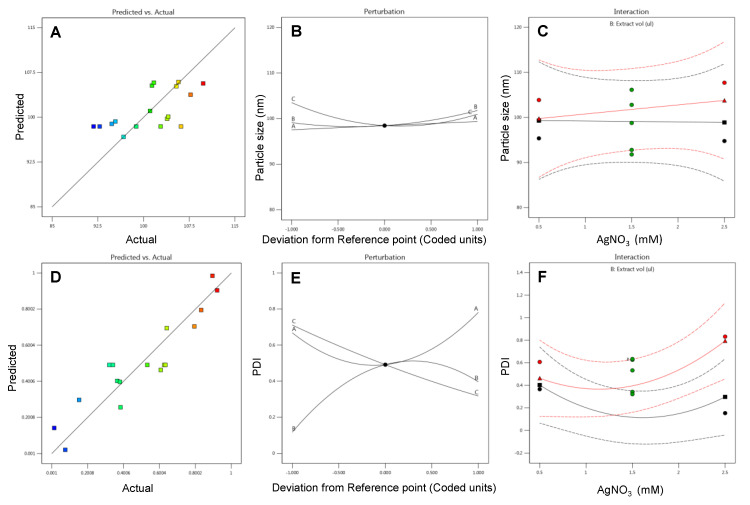
Model diagnostic plots predicted v/s actual, perturbation and interaction plots for response variables (**A**–**C**) particle size and (**D**–**F**) PDI for *P. granatum* leaves extract mediated silver nanoparticles.

**Figure 3 molecules-26-05762-f003:**
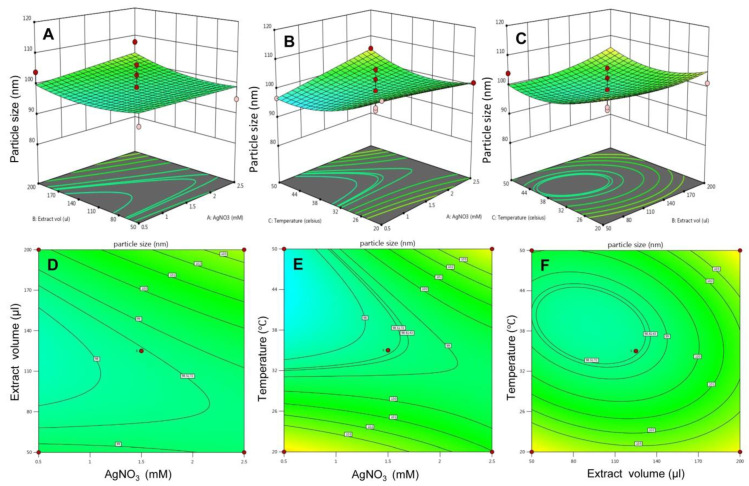
The 3D surface plot and the 2D contour plots for response—particle size. (**A**) 3D surface plot for effect of extract volume and AgNO_3_ on particles size and (**D**) is 2D plot for the same, (**B**) 3D surface plot for effect of temperature and AgNO_3_ on particles size and (**E**) is 2D plot for the same, (**C**) 3D surface plot for effect of temperature and AgNO_3_ on particles size and (**F**) is 2D plot for the same.

**Figure 4 molecules-26-05762-f004:**
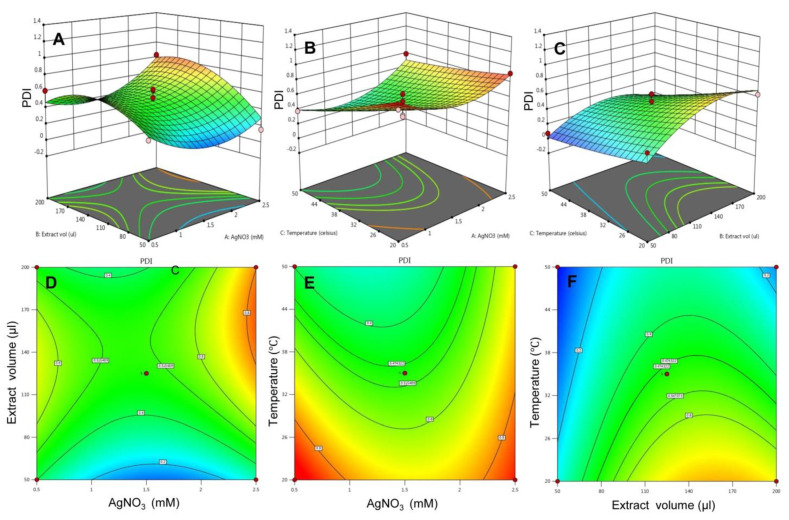
The 3D surface plot and the 2D contour plots for response—PDI. (**A**) 3D surface plot for effect of extract volume and AgNO_3_ on PDI and (**D**) is 2D plot for the same, (**B**) 3D surface plot for effect of temperature and AgNO_3_ on PDI and (**E**) is 2D plot for the same, (**C**) 3D surface plot for effect of temperature and AgNO_3_ on PDI and (**F**) is 2D plot for the same.

**Figure 5 molecules-26-05762-f005:**
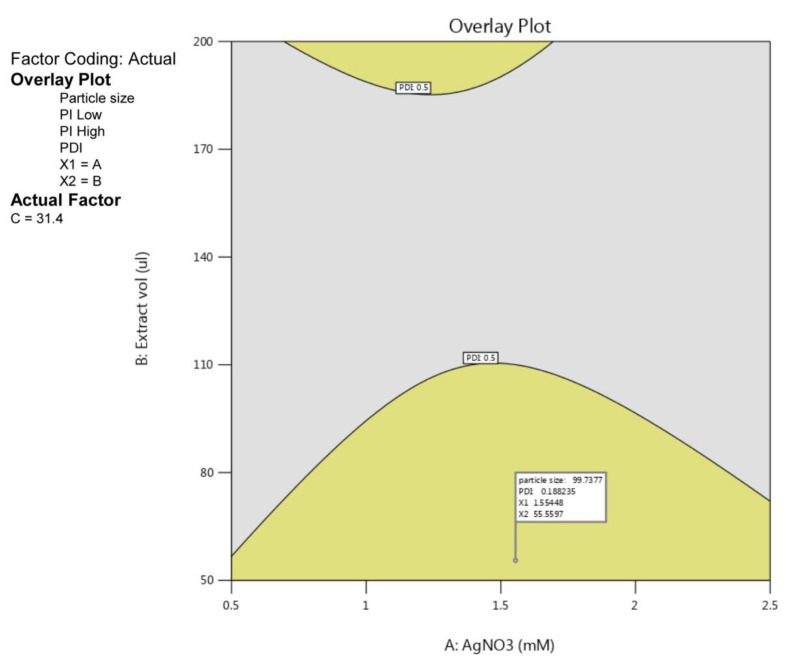
Overlay plot with a flagged point showing optimal conditions for the biosynthesis of PGL-AgNPs.

**Figure 6 molecules-26-05762-f006:**
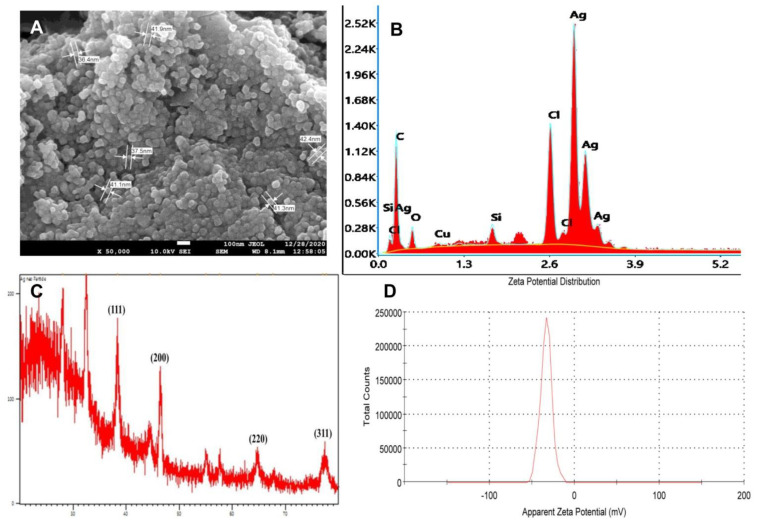
Characterization of PGL-AgNPs synthesized at optimal conditions i.e., 25 °C when the concentration of AgNO_3_ is 1.0 mM, at extract to AgNO_3_ ratio of 1:10 and the reaction continued for 15 min. (**A**) An SEM image of PGL-AgNPs with an average size of ~37.5 nm, (**B**) an EDX pattern of PGL-AgNPs, (**C**) an XRD spectrum of AgNPs, (**D**) a Zeta analysis showing zeta potential is 34 mV.

**Figure 7 molecules-26-05762-f007:**
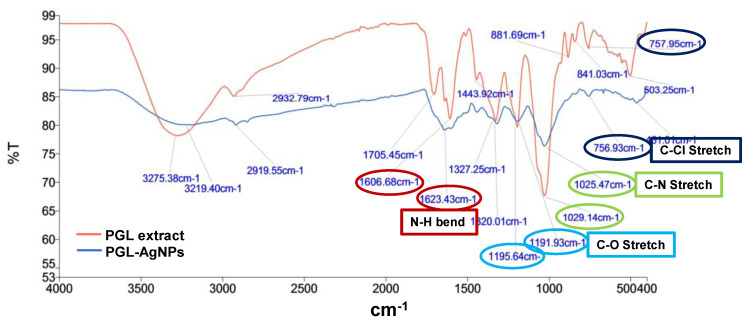
FTIR spectra of *P. granatum* leaves extract and PGL-AgNPs.

**Figure 8 molecules-26-05762-f008:**
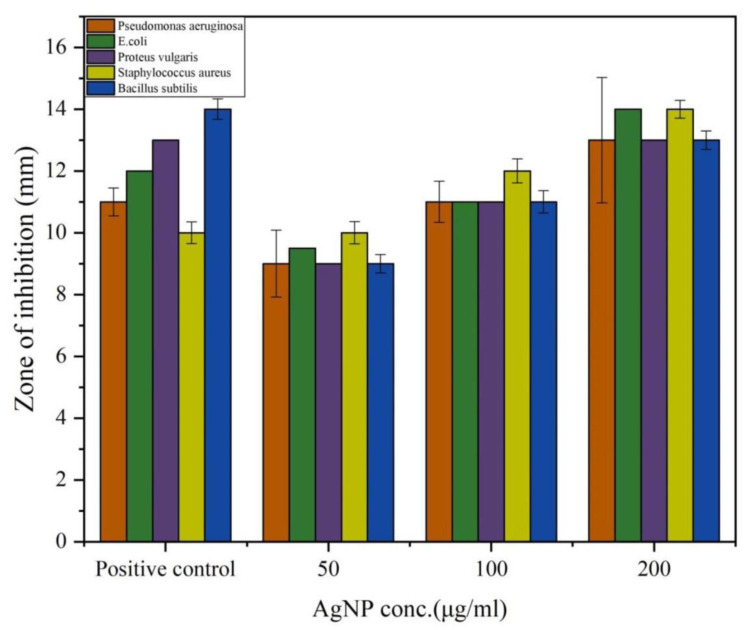
Antimicrobial activity of PGL-AgNPs by agar disc diffusion assay. Varying concentrations of AgNPs were investigated against five different bacteria. Graphical representation of zone of inhibition obtained using different concentrations of AgNPs.

**Figure 9 molecules-26-05762-f009:**
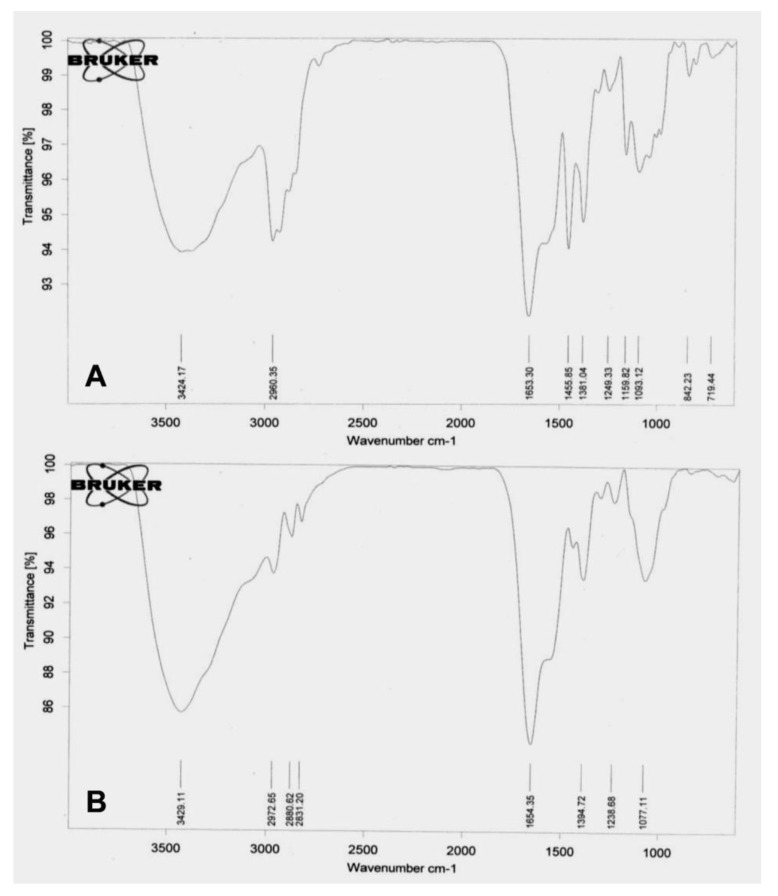
FTIR spectra of (**A**) untreated and (**B**) PGL-AgNPs treated bacterial cells.

**Figure 10 molecules-26-05762-f010:**
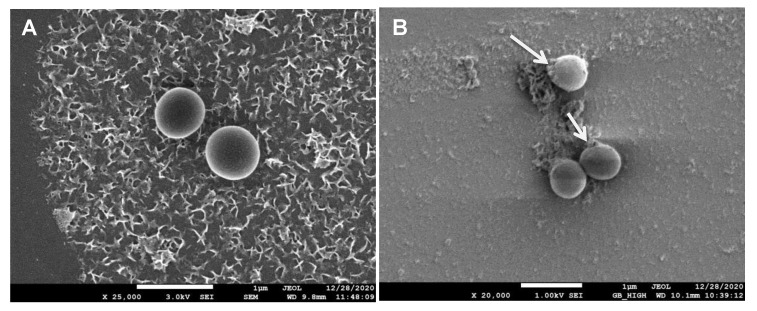
SEM image of (**A**) untreated and (**B**) PGL-AgNPs-treated bacterial cells.

**Figure 11 molecules-26-05762-f011:**
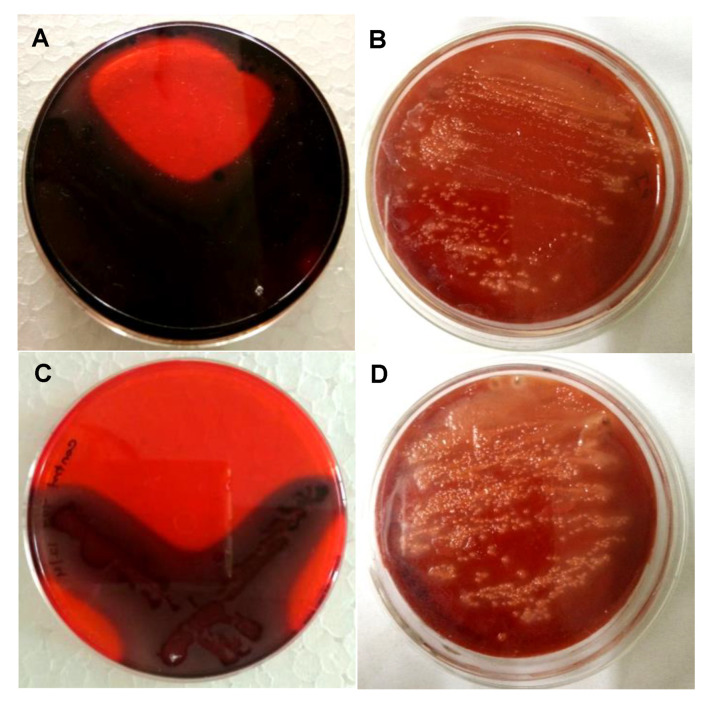
Assessment of the antibiofilm activity of PGL-AgNPs by CRA plate method. The appearance of black colonies (**A**,**C**) indicates biofilm production, while AgNPs treatment inhibits black colony formation. (**A**,**B**)—*P. aeruginosa*, (**C**,**D**)—*S. aureus*.

**Figure 12 molecules-26-05762-f012:**
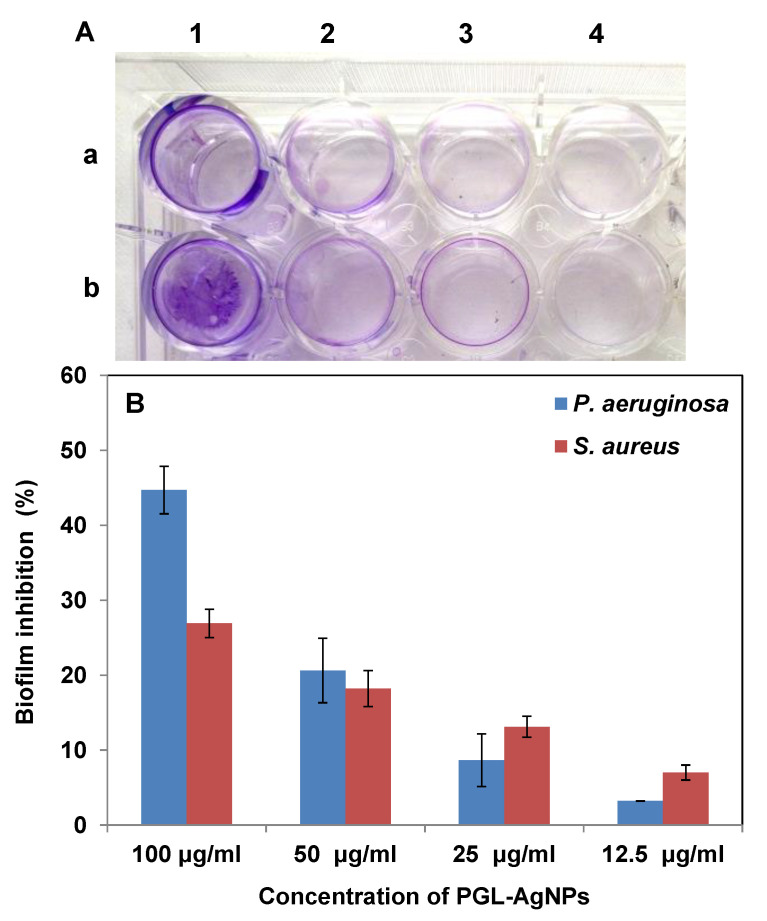
Assessment of the antibiofilm activity of PGL-AgNPs by crystal violet assay. (**A**) rows a and b represent *P. aeruginosa* and *S. aureus*, respectively, with increasing PGL-AgNPs concentrations (12.5 to 100 µg/mL) marked as 1–4, (**B**) Quantitative representation of antibiofilm activity. The biofilm inhibition was dose-dependent.

**Table 1 molecules-26-05762-t001:** Determination of MIC for PGL-AgNPs against Gram-positive and Gram-negative bacteria.

Organism	MIC (mg/mL)	Reported MIC (mg/mL)
*E. coli*	0.050 ± 0.005	0.016 [42], 0.125 [43], 0.64 [45]
*P. aeruginosa*	0.050 ± 0.002	0.032 [42], 0.065 [43], 0.016 [45]
*P. vulgaris*	0.075 ± 0.002	0.016 [43]
*B. subtilis*	0.100 ± 0	0.064 [43]
*S. aureus*	0.125 ± 0	0.128 [42], 0.125 [43], 0.625 [44], 0.256 [45]

**Table 2 molecules-26-05762-t002:** Independent variables taken for systematic optimization by the Box-Behnken design.

S. No.	Independent Variables	Units	Low Value	High Value
1	AgNO_3_ concentration	Millimolar (mM)	0.5	2.5
2	Extract to AgNO_3_ ratio	Microliter (µL)	50	200
3	Temperature	Celsius (℃)	20	50

**Table 3 molecules-26-05762-t003:** Experimental trial runs given by Box- Behnken Design.

Trial Run	AgNO_3_ Concentration (mM)	Extract to AgNO_3_ Ratio (µL)	Reaction Temperature (℃)	Particles Size (nm)	PDI
1	2.5	125	20	101.061	0.921023
2	2.5	50	35	94.7784	0.153355
3	1.5	50	50	104.04	0.0763458
4	0.5	125	50	96.7092	0.38042
5	0.5	125	20	105.727	0.894881
6	1.5	125	35	106.139	0.341596
7	2.5	125	50	105.371	0.794349
8	1.5	125	35	102.797	0.626777
9	2.5	200	35	107.711	0.832383
10	0.5	200	35	103.869	0.607419
11	1.5	200	20	101.378	0.64091
12	1.5	200	50	101.686	0.0149703
13	1.5	125	35	98.7829	0.32121
14	1.5	125	35	91.7955	0.633585
15	1.5	125	35	92.7959	0.53263
16	1.5	50	20	109.783	0.383734
17	0.5	50	35	95.368	0.36545

## Data Availability

Not applicable.

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
