# Peer review of "Exploring the Antibacterial and Antibiofilm Efficacy of Silver Nanoparticles Biosynthesized Using Punica granatum Leaves"

_molecules, 2021, doi:10.3390/molecules26195762_

Round 1

Reviewer 1 Report

The manuscript describes the biosynthesis of AgNPs using leaves extract of Punica granatum investigating the effect of these nanoparticles as antibacterial and antibiofilm. The work is good, but I have some comments:

  • The resolution of photos and diagrams included are very poor and must be replaced or justified
  • the manuscript needs further revision of writing (please find the attached manuscript)
  • Please make sure that you have detailed the abbreviations in the whole manuscript

Author Response

We would like to thank the editors for reviewing our manuscript. We have carefully gone through the comments/suggestions and made the necessary changes in the revised manuscript. We sincerely hope that the revision has brought the manuscript to the level of reviewer’s satisfaction. Please find below our point-by-point reply to the comments and the modifications incorporated in the manuscript thereafter. All the changes in the manuscript are highlighted with track changes.

Response to the comments

Referee 2

Comment 1: The resolution of photos and diagrams included are very poor and must be replaced or justified

Reply: As per the suggestion resolution of photos and diagrams have been adjusted up to the mark.

Comment 2: the manuscript needs further revision of writing (please find the attached manuscript)

Reply: As per the suggestion of the reviewer we have thoroughly gone through the manuscript and it has been revised.

Comment 3: Please make sure that you have detailed the abbreviations in the whole manuscript

Reply: The manuscript has been rechecked to make sure that all abbreviations have been detailed.

Reviewer 2 Report

This article presents the antibacterial and antibiofilm efficacy of silver nanoparticles biosynthesized using Punica granatum leaves. All conclusions are supported by the experimental data and the experiments were well conducted. I find the result of this work meaningful. This work will be suitable for publication after addressing the following point:

  1. Please revised the figure 6B, which is not clear for reading.
  2. In figure 6C, it is better to insert the data of JCPDS file No. 04-0783 as comparison.
  3. Could you explain how and why AgNPs treat Gram negative and Gram positive bacteria differently. And please describe the positive control in figure 8.
  4. Please discuss the figure 12A. It seems no related explanations.
  5. Please explain why runs 17 experimental trials and what its conclusion is.
  6. The manuscript contains a few grammar/typographical/etc errors. Please check it up carefully.

Author Response

We would like to thank the editors for reviewing our manuscript. We have carefully gone through the comments/suggestions and made the necessary changes in the revised manuscript. We sincerely hope that the revision has brought the manuscript to the level of reviewer’s satisfaction. Please find below our point-by-point reply to the comments and the modifications incorporated in the manuscript thereafter. All the changes in the manuscript are highlighted with track changes.

Response to the comments

Referee 3

Comment 1: Please revise the figure 6B, which is not clear for reading.

Reply: As per the reviewer’s suggestion we have revised figure 6B and image with better resolution has been uploaded in the manuscript.

Comment 2: In figure 6C, it is better to insert the data of JCPDS file No. 04-0783 as comparison.

Reply: We welcome the suggestion of the reviewer, however, the JCPDS file no. 04-0783 is very common and freely available, hence we didn’t include that in figure 6C, just to avoid overcrowding and  reuse the data that is freely available.

Comment 3: Could you explain how and why AgNPs treat Gram negative and Gram positive bacteria differently. And please describe the positive control in figure 8.

Reply: The reason why AgNPs treat Gram negative and Gram positive bacteria differently have been briefed in the manuscript under the section 2.8.

Comment 4: Please discuss the figure 12A. It seems no related explanations.

Reply: As per reviewer’s suggestion figure 12 A has been explained under the section 2.9 paragraph 2.

Comment 5: Please explain why 17 experimental trials runs and what its conclusion is.

Reply: Box-Behnken Design (BBD) is a tool for experimental design and parameter optimizations. It is a technique to determine the impact of different factors (independent variables) on responses (dependent variables). To perform BBD, identification of independent variables is a preliminary step which is essential for biosynthesis process. In BBD, a minimum of three factors were required and each factor always include three levels. For three factors three blocks are required where other two factors vary keeping one factor as constant. This can be explained by imagining a cube (also referred as design space) where midpoints (rotatable) of each edge represent combinations of different factors. It is also mandatory to involve centre point of the cube at which all the factors are at their mid values. In the experimental design, some experimental or trial runs were suggested by BBD where each run is a mixture of different levels of factors. The number of trial runs based upon the given parameters was by default suggested by the tool itself.  Each suggested trial run were then investigated to determine the influence of independent variables on each response.

Comment 6: The manuscript contains a few grammar/typographical/etc errors. Please check it up carefully.

Reply: The manuscript has been thoroughly checked and the grammatical errors rectified.

Reviewer 3 Report

The manuscript can be accepted for publication in Molecules, after minor revision. The authors should revise the manuscript according to the following comments.

  1. the authors should upload the figures of petri with the inhibition zones in the supplementary materials
  2. the authors should give a table with the IZ and MIC values with the confidence limits
  3. the authors mention that “Figure 9 A and B illustrates the difference between membranes of PGL-AgNPs treated S. aureus along with proper control.” However in the figure capture “FTIR spectra of untreated and PGL-AgNPs treated bacterial cells” please check it accordingly.
  4.  please give information what figure 12 mentioned, what is 1-4, a-b?
  1. The authors should also calculate the BEC value of the nanoparticles

Author Response

We would like to thank the editors for reviewing our manuscript. We have carefully gone through the comments/suggestions and made the necessary changes in the revised manuscript. We sincerely hope that the revision has brought the manuscript to the level of reviewer’s satisfaction. Please find below our point-by-point reply to the comments and the modifications incorporated in the manuscript thereafter. All the changes in the manuscript are highlighted with track changes.

Response to the comments

Referee 1

Comment 1: the authors should upload the figures of petri with the inhibition zones in the supplementary materials

Reply: As per the reviewer’s suggestion the figures of petri plates showing zone of inhibition have been uploaded in supplementary materials as figure S1.

Comment 2: the authors should give a table with the IZ and MIC values with the confidence limits

Reply:  As per the suggestion tables showing zone of inhibition and MIC values along with standard deviation have been modified and inserted in the manuscript.

Comment 3: the authors mention that “Figure 9 A and B illustrates the difference between membranes of PGL-AgNPs treated S. aureus along with proper control.” However in the figure capture “FTIR spectra of untreated and PGL-AgNPs treated bacterial cells” please check it accordingly.

Reply: As per the suggestion the content has been revised under the section 2.8 and necessary corrections have been made.

Comment 4: please give information what figure 12 mentioned, what is 1-4, a-b?

Reply: As per the suggestion labels in figure 12 were briefed in figure legend.

Comment 5: The authors should also calculate the BEC value of the nanoparticles

Reply: We did not calculate BEC value of PGL-AgNPs which determine the mass concentration of silver instead we simply estimated AgNPs in the dried form by using four digit weighing balance post biosynthesis. Secondly, the outsourcing facilities for inductively coupled plasma mass spectrometry (ICP-MS) was not easily available.

Round 2

Reviewer 2 Report

I happy with the revised manuscript. Just one more suggestion, it has to remove the instrumental logo in the figure 9.